# The mutREAD method detects mutational signatures from low quantities of cancer DNA

Juliane Perner[1,4], Sujath Abbas [2,4], Karol Nowicki-Osuch [2,4], Ginny Devonshire [1], Matthew D. Eldridge [1], Simon Tavaré[1,3] & Rebecca C. Fitzgerald [2✉]

Mutational processes acting on cancer genomes can be traced by investigating mutational signatures. Because high sequencing costs limit current studies to small numbers of good-quality samples, we propose a robust, cost- and time-effective method, called mutREAD, to detect mutational signatures from small quantities of DNA, including degraded samples. We show that mutREAD recapitulates mutational signatures identified by whole genome sequencing, and will ultimately allow the study of mutational signatures in larger cohorts and, by compatibility with formalin-fixed paraffin-embedded samples, in clinical settings.

[1] Cancer Research UK Cambridge Institute, University of Cambridge, Li Ka Shing Centre, Cambridge, UK. [2] Medical Research Council Cancer Unit, Hutchison/ Medical Research Council Research Centre, University of Cambridge, Cambridge, UK. [3] Irving Institute for Cancer Dynamics, Columbia University, New York, NY, USA. [4] These authors contributed equally: Juliane Perner, Sujath Abbas, Karol Nowicki-Osuch. ✉email: RCF29@MRC-CU.cam.ac.uk

Genome instability is a hallmark of many cancers and leads to the accumulation of single nucleotide variants and copy number alterations in tumor cells. The analysis of the prevalence of specific nucleotide substitutions throughout the genome has revealed that mutational processes, to which the cells are exposed, leave footprints, termed mutational signatures[1–3]. Large-scale genome sequencing efforts on different cancer types have identified over 50 mutational signatures[2,4,5] and their detailed characterization has improved our understanding of the cellular defects acting on cancer genomes[6–9] and their evolution in normal tissues[10–12]. Recent studies have shown that the mutational signatures can be used for patient stratification, for example, to help tailor therapies to exploit specific defects in these patient sub-groups or to improve early detection and cancer prevention strategies[13–18].

Mutational signatures in a tumor genome usually have been derived from whole-genome sequencing (WGS)[4,5,19,20]. Due to the associated sequencing costs, WGS is generally limited to studies with small numbers of high-quality samples. The successful application of mutational signatures in clinical settings will thus depend on the availability of a cost-effective, scalable detection method that can handle samples of low quality containing small amounts of DNA.

The relative contribution of mutational processes to the overall mutational spectrum in DNA samples is deciphered mathematically from the frequency of substitutions in their trinucleotide context. Under the assumption that the frequencies can be accurately estimated from a subset of mutations, sequencing at lower genome-wide coverage, i.e. shallow WGS at 10× coverage (10× sWGS)[13], and whole-exome sequencing (WES) have been proposed as potential alternatives to WGS for detecting mutational signatures. However, the low coverage of 10× sWGS can lead to spurious mutations calls and will likely bias the detected mutations to those highly abundant in the cell population of the DNA sample. On the other hand, WES masks the contribution of intergenic mutations to the mutational spectrum, potentially leading to a biased estimation of the presence of mutational signatures.

Here, we propose an easy-to-use method for mutational signature detection building on reduced representation sequencing (RR-seq) approaches that have been successfully applied in population genetics analyses[21,22]. Our protocol is based on sequencing a reproducible, random subset of genomic regions generated by double-enzymatic digestion and subsequent fragment size-selection of the DNA sample. As a result, sufficient coverage for somatic mutation calling is achieved without bias in the type of detected mutations. The proposed method can detect mutational signatures from small quantities of DNA, including degraded samples from formalin-fixed paraffin-embedded (FFPE) material, in a robust, cost- and time-effective manner.

## Results

**Performance of mutREAD in computational simulations**. Our proposal assumes that obtaining a random subset of all mutations is sufficient to determine the presence of mutational signatures. To test this assumption, we first performed computational simulations (see "Methods" section) using available data from whole-genome sequencing of 129 esophageal adenocarcinoma (EAC) samples and the six mutational signatures derived from them[13].

The stability of the mutational signature profile was evaluated as a function of the number of randomly selected mutations detected in the WGS samples (Fig. 1a). The cosine similarity relative to the original mutational signature profile increases with the number of mutations available for estimation. A plateau is reached at 500 mutations, suggesting that fewer than the WGS-derived number of mutations (on average 26k mutations per EAC sample) are sufficient to obtain the mutational signature profile.

The second assumption is that the mutation subset generated by RR-seq is an unbiased representation of the mutational spectrum. We simulated subsets of mutations for RR-seq using different enzyme combinations, as well as for 10× sWGS and WES (see "Methods" section). In this simulation, RR-seq with at least 161 out of 169 enzyme combinations outperforms (expanded) WES and 10× sWGS in terms of average cosine similarity between the WGS-derived and simulated signature profile in EAC (Fig. 1b). This difference can in part be attributed to the number of mutations recovered by the different methods (WES: 211, expanded WES: 282, 10× sWGS: 462 and RR-seq: 381 mutations on average). Notably, RR-seq derived mutations originate from a much lower proportion of the genome (a range of 0.2–82 Mbps, mean: 10 Mbps, 0.3% of WGS) than (expanded) WES-based mutations (WES: 46 Mbps/1.39% of WGS; expanded WES: 62 Mbps/1.88% of WGS).

We further investigated the applicability of RR-seq for estimating mutational signatures in different cancer types using the WGS data collected by the Pan-Cancer Analysis of Whole Genomes (PCAWG) network[2]. RR-seq accurately estimated the mutational signature profiles across the majority of the 20 cancer types, including cancers with highly diverse mutational signature content, e.g. liver hepatocellular carcinoma (Liver HCC), and a non-solid tumor, i.e. B-cell non-Hodgkin lymphoma (Lymph-BNHL, Supplementary Fig. 1A). As expected from our simulations above, the performance of the method was correlated with the mutational load across cancer types (Supplementary Fig. 1B). Finally, RR-seq outperformed (expanded) WES in all cancer types (Supplementary Fig. 1C).

**Implementation and optimization of mutREAD**. Having established the superiority of RR-seq over other methods in the simulation, we implemented our approach, which we called mutREAD (mutational signature detection by restriction enzyme-associated DNA sequencing), by adapting and improving on the principles of the quaddRAD protocol[21]. Key features of the protocol include incorporation of unique molecular identifiers (UMI) and inline barcodes, which allow for computational identification of PCR duplicates and larger multiplexing capabilities, respectively (Fig. 1c). The protocol is further streamlined by simultaneous enzymatic digestion and adapter ligation and removal of unnecessary purification steps. Here, we optimized the protocol towards application to EAC, for which six mutational signatures have been previously identified from WGS on fresh-frozen samples[13]. In particular, we chose the optimal pair of enzymes based on the simulation described above. The enzyme combination PstI and ApoI showed one of the highest cosine similarities to WGS results in EAC (Fig. 1b), as well as broad genome coverage and even distribution of target loci throughout the genome (Supplementary Fig. 2). Hence, we designed adapter sequences that terminated with PstI and ApoI restriction enzyme compatible sites and that are devoid of PstI or ApoI restriction enzyme sites to avoid digestion of the adapters (Supplementary Table 1).

We further optimized the protocol to suit either fresh-frozen or FFPE samples (see "Methods" section), the latter being the standard sample preservation strategy in clinical practice. Restriction enzyme double digestion, adapter ligation conditions, and size selection were optimized for optimal digestion, adapter annealing, and size selection using an EAC cell line (FLO-1). The protocol was further adjusted for FFPE derived DNA from the same EAC cell line (Supplementary Fig. 3).

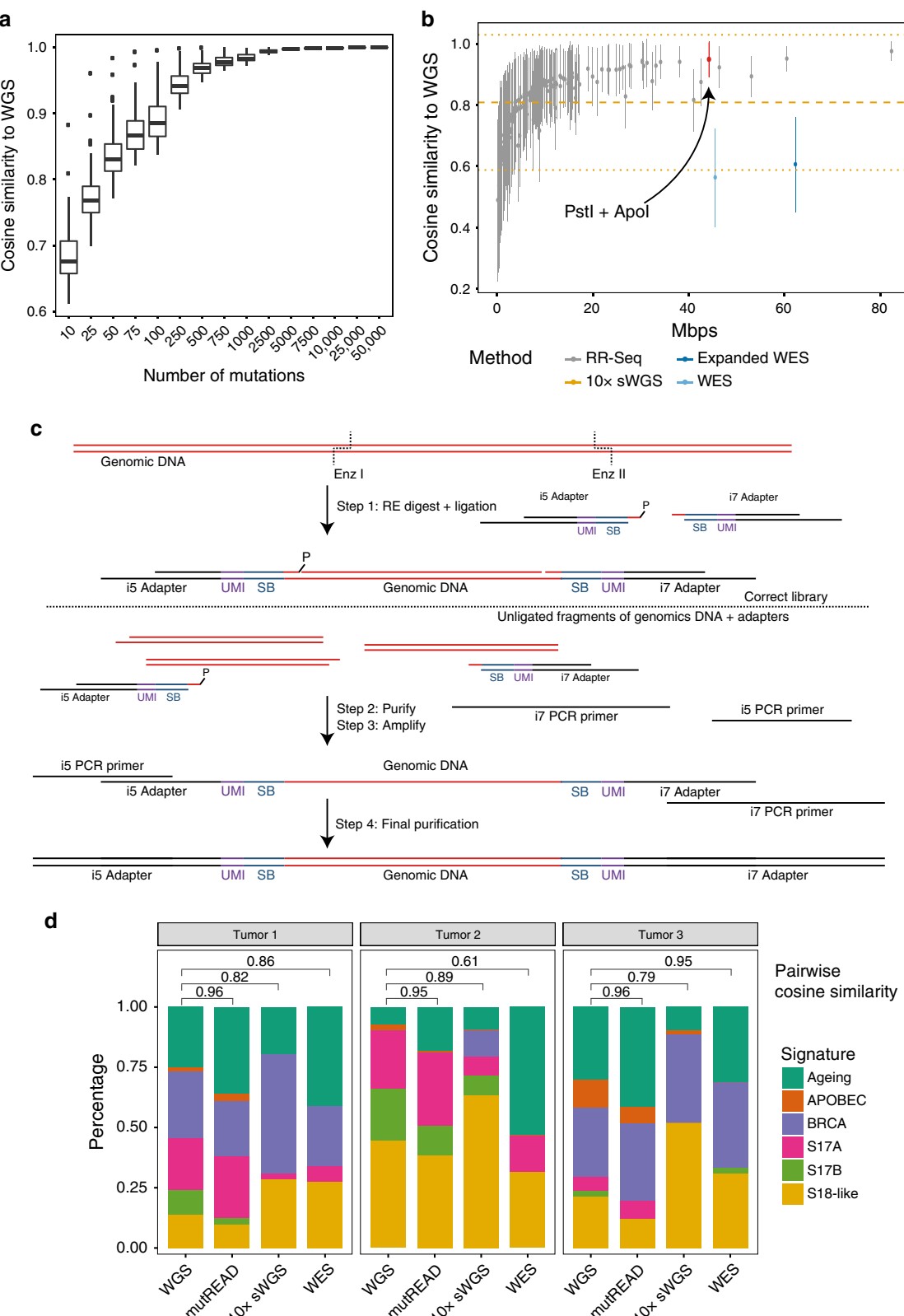

**mutREAD outperforms other approaches for calling muta-tional signatures**. We then applied mutREAD to fresh-frozen tumor and matched blood samples from biopsies of three different EAC patients and evaluated the quality of the library under several criteria (Supplementary Table 2, Supplementary Fig. 4). The mutational signatures, derived from 530 to 1471 mutations detected using GATK Mutect2[23], showed cosine similarities of 0.95–0.96 when compared with the WGS-derived mutational signature profiles (Fig. 1d). We observed similar cosine similarity between mutREAD and WGS when mutations were derived using an alternative mutation caller, Strelka[24] (Supplementary Table 3). We note that two mutation callers agree in 41–85% of mutations,

**Fig. 1 Comparative analysis of mutational signatures and method overview. a** Cosine similarity (*y*-axis) of whole-genome sequencing (WGS)-derived mutational signatures for EAC samples (*n* = 129 independent patients) and signatures derived from random subsets of mutations with increasing number of mutations (*x*-axis). Data are shown as boxplots, where the bold line at the center indicates the median and the upper and lower hinges extend to the 25th and 75th percentile, respectively. The upper/lower whisker extends from the upper/lower hinge to the largest/smallest value no further than 1.5 times the interquartile range from the upper/lower hinge. Samples outside this range are indicated as points. Only samples having sufficient number of mutations (at least the number indicated on the *x*-axis) contribute to the boxes. **b** Cosine similarity (*y*-axis) of WGS-derived mutational signatures for EAC samples (*n* = 129 independent patients) and signatures derived from subsets of mutations simulating different sequencing approaches (*x*-axis). Data are represented as mean cosine similarity values ± standard deviation. Different enzyme combinations were simulated for RR-seq, each shown as a different point. For the simulated 10x sWGS samples, the mean (*n* = 21 independent patients) is given as dashed horizontal line and the standard deviation is given as dotted line. RR-Seq, reduced representation sequencing; 10× sWGS, 10× shallow whole-genome sequencing; WES, whole-exome sequencing; expanded WES, whole-exome sequencing expanded to untranslated regions and miRNAs. **c** Schematic overview of the individual steps in mutREAD. Details for each step are given in the "Methods" section. SB, sample barcode; UMI, unique molecular identifier; RE, restriction enzyme. **d** Comparison of the mutational signature profiles for three EAC samples across different sequencing methods (*x*-axis). Each bar indicates the contribution of the mutational signature (*y*-axis) to the overall mutational spectrum. Pairwise cosine similarities to WGS for mutREAD, WES and 10× sWGS are indicated above the bars.

suggesting that for optimal sensitivity dedicated benchmarking and optimization is necessary when applying different mutation callers. In summary, the mutREAD protocol results in reproducible, good quality, target-specific libraries from which mutational signatures can be successfully derived.

Next, we compared mutREAD with WES and 10x sWGS libraries of the same samples sequenced to similar depth. Quality measures for the resulting libraries of the different methods are summarized in Supplementary Table 4. WES resulted in 46–325 mutations per sample and 10x sWGS identified 21–83 mutations per sample. mutREAD consistently achieved high cosine similarity to the corresponding WGS-derived signatures. Conversely, WES and 10x sWGS had lower cosine similarities and much higher variability between patients (Fig. 1d).

**High performance of mutREAD on FFPE-derived samples**. Finally, we investigated if mutREAD can be used to study historical samples by sequencing FFPE specimens matching the previously analyzed frozen samples. Fresh frozen and FFPE-derived samples generated similar signature patterns (Fig. 2a), despite the lower sequencing depth and smaller fragment distribution of final FFPE-derived libraries (Supplementary Fig. 4, Supplementary Table 2). Cosine similarities to WGS-derived mutational signatures were between 0.89 and 0.96 based on 47–383 detected mutations.

We replicated the good cosine similarity to WGS-derived mutational signatures in additional nine FFPE samples (Fig. 2b). Of note, samples were derived from tumor resections and pathology estimates for these samples show low tumor content (10–70%, Supplementary Table 5), explaining the lower number of mutations and higher variability across samples compared to the previously tested biopsy samples. Given the high degradation expected in FFPE samples which can result in variability, we also tested the reproducibility of FFPE-derived mutREAD libraries. Technical replicates of the nine FFPE samples showed high concordance in sequenced regions and fragment size distribution (Fig. 2c, Supplementary Fig. 5). Hence, while it is expected that the performance on FFPE is lower compared to fresh-frozen samples, our results suggest that mutREAD can also be applied to FFPE-derived DNA samples with low tumor content and leads to reproducible results.

## Discussion

We have presented the development and application of a cost-effective and scalable method for the detection of mutational signatures in DNA samples. mutREAD produces reproducible and highly specific reduced representation libraries and the derived mutational signatures mirror the WGS-derived signatures

with high cosine similarity. Importantly, this also holds true even when used with highly degraded DNA samples. Our method will ultimately allow the study of mutational signatures in much larger cohorts and in clinical settings where FFPE-derived DNA samples are routinely collected.

Applied to tumor samples from EAC patients, we showed that mutREAD outperforms the previously proposed methods WES and 10× sWGS. EAC is characterized by abundant somatic mutations, which are most prevalent in intergenic and intronic regions[13,25]. The choice of library preparation methods to study mutational signatures in other cancer types will depend on the overall mutation rate and the genomic distribution of the somatic mutations.

In terms of scalability and cost mutREAD outperforms other methods (Supplementary Table 6). In our hands, the cost associated with mutREAD libraries synthesis is 80% lower than for 10× sWGS and 96% lower than for WES libraries. Sequencing costs on the Illumina HiSeq 4000 are comparable for WES and mutREAD libraries, while sequencing 10× WGS libraries is at least three times more expensive. Further, due to its high multiplexing capabilities for sequencing and for library preparation mutREAD is highly scalable for studying larger cohorts.

Given its ease of use and low cost, we envision a wide range of applications for mutREAD to study mutational signatures in basic research and translational settings. For example, clinical trials using mutational signature-based patient stratification to assign optimal therapies become feasible. mutREAD could further improve the mutational signature-based prediction of homologous recombination deficiency in clinical samples[14,26]. Together with computational tools for coarse-grained copy alteration detection[22,27], mutREAD could provide a detailed view of the role of mutational processes in cancer progression and evolution from archived material. Finally, correlative analyses of mutational signatures with endogenous and environmental parameters to understand the source of so far unknown mutational signatures will shed light on the etiology of cancers.

## Methods

**Enzyme selection criteria**. The enzyme combination is an important parameter to optimize for the mutREAD method. We focused on high-fidelity restriction enzymes provided by New England BioLabs Inc. (Ipswich, Massachusetts USA) to allow for fast DNA digestion and maximum target specificity under a broad range of experimental conditions. Since cancer samples frequently exhibit DNA hyper- or hypo-methylation, which could affect restriction enzyme sites, we required insensitivity to CpG methylation status. To simplify the adapter design, only enzymes with a unique cut-site including only A, C, G, and T were considered. Finally, cut sites were required to have a maximum length of six base pairs to increase the number of generated fragments. The tested list of enzymes is given in Supplementary Table 7.

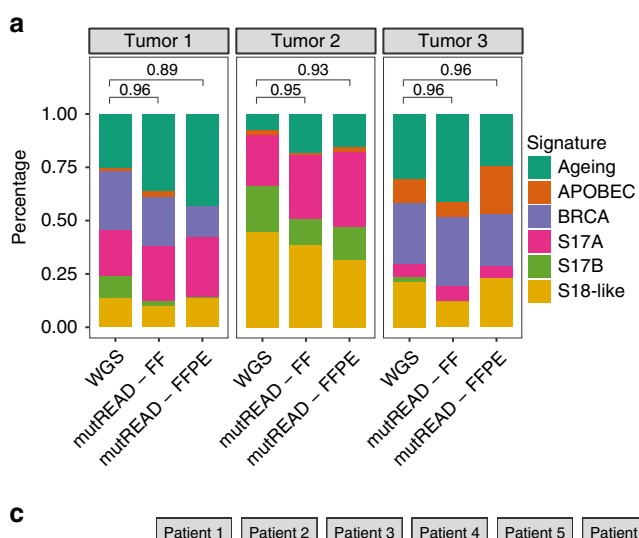

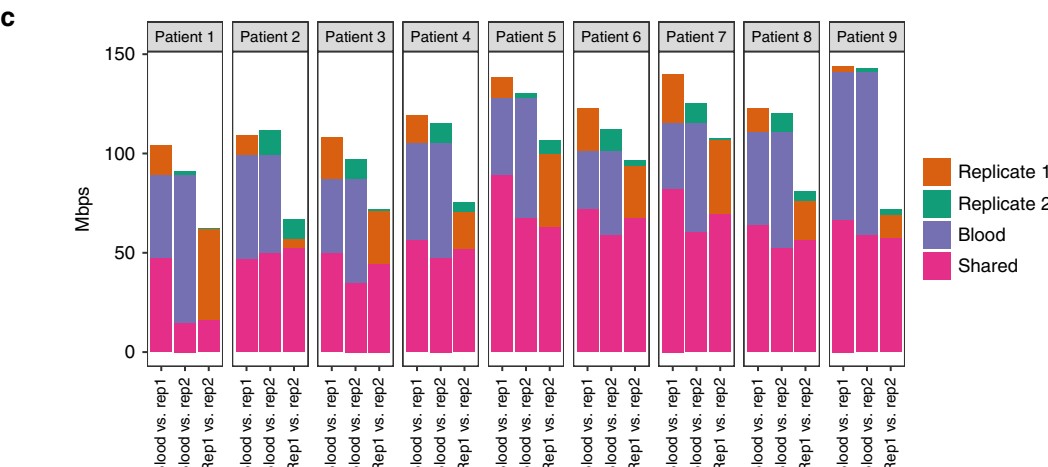

**Fig. 2 mutREAD reproducibly detects mutational signatures in FFPE samples. a** Comparison of the mutational signature profiles between WGS, fresh-frozen (FF) and FFPE samples for the same three EAC samples as in Fig. 1. Each bar indicates the contribution of the mutational signature (y-axis) to the overall mutational spectrum. Pairwise cosine similarities to WGS for the two mutREAD libraries are indicated above the bars. **b** Cosine similarity between mutational signatures derived from nine additional FFPE and WGS sample pairs and the number of detected mutations in the FFPE samples used to derive the mutational signatures. **c** Reproducibility of the sequenced regions between the first FFPE-derived technical replicate and the blood sample, the second FFPE-derived technical replicate and the blood sample, and between the two technical replicates (x-axis). The bars indicate the size of the overlapping regions in million base pairs (Mpbs, y-axis) for each comparison. Only regions covered at least 10× contribute to the comparison. The second technical replicate was sequenced to lower coverage and we down-sampled the first technical replicate by 50% to approximately match the sequencing coverage for comparison.

**Simulations**. We opted for a double-digest protocol to produce fragments that are reproducible between libraries. To simulate the performance of all possible enzyme combinations fulfilling the above criteria, we use ddRADseqTools (v0.45)[28] to perform in silico digestion of the human hg19 reference genome and size selection for fragments of expected length between 350–450 bp. The expected fragment size range of 350–450 base pairs was chosen as the maximum fragment size such that the complete library fragments (insert, adapters and primers) could still be sequenced on a standard Illumina HiSeq system. WGS-based mutations were selected if they overlap the resulting expected fragments and mutational signatures were calculated based on this selection. Similarly, WES and expanded WES sequencing is simulated using the target regions provided by Nextera for the rapid capture exome/expanded exome kit (v1.2)[29], where the exome kit comprises 45 Mbps of coding regions and the expanded exome kit comprises 62 Mbps of coding regions, untranslated regions and miRNAs. Further, the 21 simulated 10x sWGS libraries from a previous study[13] were used. In short, the 10x sWGS were simulated by down-sampling the WGS libraries and re-running the mutational calling.

**Cosine similarity**. We measure similarity between two mutational signature profiles **P** and **Q** using the cosine similarity. The cosine similarity between the non-zero vectors **P** and **Q** with n mutational signatures is defined as

$cossim(\mathbf{P}, \mathbf{Q}) = \frac{\sum_{i=1}^{n} \mathbf{P}_i \mathbf{Q}_i}{\sqrt{\sum_{i=1}^{n} \mathbf{P}_i^2} \sqrt{\sum_{i=1}^{n} \mathbf{Q}_i^2}}$. Two mutational signature profiles that are

independent have cosine similarity of 0. Conversely, identical mutational signature profiles obtain a cosine similarity of 1.

**Computational simulations using Pan-cancer analysis of whole-genomes data**. We also performed computational simulations on the WGS data from the PCAWG network. The collection was downloaded from https://dcc.icgc.org/releases/PCAWG/consensus_snv_indel. We have used the signature compendium from COSMIC (v3, downloaded from https://dcc.icgc.org/releases/PCAWG/mutational_signatures/Signatures/SP_Signatures/SigProfiler_reference_signatures) to capture all mutational signatures relevant to the different cancer types. Only cancer types with at least 10 samples present in the collection were analyzed.

**Ethical approval, sample collection**. Esophageal adenocarcinoma samples were collected by the Oesophageal Cancer Classification and Molecular Stratification (OCCAMS) project, a multi-center UK-wide study. The study was approved by the Institutional ethics committee (REC 07/H0305/52 and 10/H0305/1) and included individual informed consent.

**Assay optimization**. All optimization experiments were performed using 500 ng of genomic DNA from an EAC cell line (FLO-1), commercially available from culture collection of Public Health England. In-house STR analysis was done in the lab to confirm a >90% match prior to assay optimization. Experiments were then repeated with frozen tumor, matched blood and FFPE tumor DNA from EAC patients.

**DNA extraction and quantification**. DNA was extracted from FLO-1 cell line and frozen tumors using the Allprep DNA/RNA mini kit (Qiagen, Hilden Germany) and DNA from blood was isolated using QIAmp DNA blood maxi kit (Qiagen,

Hilden Germany). AllPrep DNA/RNA FFPE Kit (Qiagen, Hilden Germany) was used to extract DNA from FFPE tumors. DNA quantification was done using Qubit dsDNA Broad Range (BR) assay kit on Qubit 3.0 fluorometer (Thermo Fisher Scientific, Waltham Massachusetts USA).

**Restriction digestion optimization for ApoI HF-PstI HF double digest.** High-fidelity (HF) ApoI and PstI restriction enzymes were obtained from New England BioLabs Inc. (Ipswich, Massachusetts USA). The optimization of restriction enzyme digestion (Supplementary Fig. 4) was performed on 500 ng of FLO1 cell line genomic DNA and included optimization of enzyme concentration, library purification procedure, PCR cycle optimization and removal of FFPE artefacts.

**Adapter design and primers.** Adapters (i5 and i7, Supplementary Table 1) were designed to target DNA fragments with restriction overhangs for the selected restriction enzymes (PstI and ApoI) and achieve specific and uniform sampling of the genome by modifying Illumina adapter sequences[30] following the general principles of the quaddRAD protocol[21]. The random 4 bp degenerate barcode included in both, i5 and i7, was designed to avoid creating new restriction sites. The 6 bp unique inner barcode sequences were balanced for A/C and G/T content to increase the sequence diversity at each position across the inner barcodes. Additionally, PhiX control was spiked in to 20% to improve the overall sequencing quality. The i5 upper adapter was phosphorylated to abolish the ligation at the 3' end and the lower i5 adapter was phosphorylated for its ligation with the DNA insert. To avoid non-specific amplification during the PCR stage the i7 adapters were designed in a Y-shape conformation to amplify only those DNA fragments with specific adapters ligated to them. Illumina universal PCR primers (i5nn and i7nn) were used for amplification (Supplementary Table 1). A phosphorothioate bond at the 3' end of the outer barcodes/primers (i5nn/i7nn) was added to protect from nonspecific or proofreading nuclease degradation.

**Adapter preparation.** Lyophilized adapters obtained from Integrated DNA Technologies (IDT, Leuven Belgium) were reconstituted in Tris-EDTA (TE pH:8) buffer to get 100 μM stock. Complementary upper and lower single strands of i5 and i7 were annealed at 10 μM each using annealing buffer (500 mM NaCl,100 mM Tris-HCl, pH 7.5–8) on a thermal cycler with the following conditions: Denature at 97.5 °C for 2.5 min and then bring down to 4 °C at a rate of 3 °C/min. Hold at 4 °C. Adapters were stored in −20 °C. This 10 μM working dilution of adapters stock was used in the ligation reaction.

**Library preparation and sequencing.** Both restriction digestion and ligation reaction were performed simultaneously. 500 ng of genomic DNA was digested with 50 U of PstI-HF and ApoI-HF in presence of 0.187 mM mutREAD i5 and i7 adapters, 400 U of T4 ligase and 1 mM ATP in 1X CutSmart buffer. The reaction was incubated on a thermal cycler at 30 °C for 3 h. Ligation reaction was stopped by addition of 10 μl of 50 mM EDTA.

Two step size selection for 400–500 bp inserts (DNA fragments, excluding adapters) was performed using Agencourt AMPure XP beads (BECKMAN COULTER, Brea California US). Unwanted larger fragments were removed with 0.6× ratio of AMPure beads to ligation product and the short fragments were removed by 0.15× size selection.

The size selected DNA fragments ligated with adapters (20 μl) were amplified using PCR primers (i5nn/i7nn) compatible with Illumina sequencing platform. The reaction was performed in total volume of 100 μl with 0.8 U of Phusion high-fidelity polymerase, in the presence of 0.2 mM dNTPs and 1X Phusion High Fidelity buffer. PCR was performed in the following conditions: 98 °C/2 min denaturation, 12 cycles of amplification at 98 °C/10 s, 65 °C/30 s, 72 °C/30 s and final extension at 72 °C for 5 min. Libraries were purified using 0.8X AMPure beads (80 μl beads + 100 μl library), this step was repeated one more time to remove all unwanted leftover reactants during PCR. Libraries were eluted in 20 μl TE buffer (Tris-EDTA buffer 10 mM Tris-HCl and 0.1 mM EDTA, pH 8) and stored at −20 °C. Quality control was performed on Agilent 2100 Bioanalyzer using Agilent High Sensitivity DNA kit (Santa Clara, California, US) or High Sensitivity D1000 TapeStation kit (Agilent). Quantification of the libraries was performed using KAPA Library Quantification kit (KK4953-07960573001 for Illumina platforms, Kapa Biosysytems Roche Holding AG Basel Switzerland) on the Light cycler 480 (Roche Life Sciences, Basel Switzerland). Libraries with unique adapters were pooled and sequenced on the HiSeq4000 using paired end, 150 bps chemistry.

**De-multiplexing and PCR duplicate identification.** After sequencing, all libraries were de-multiplexed using the outer barcodes. Next, for libraries containing random/degenerated molecular barcodes, PCR duplicates were identified and removed using Stacks' clone_filter (version 1.46)[31], allowing for random oligos of length 4 bp at both ends of the read pair. Another round of de-multiplexing using all possible combinations of inner barcodes, low quality read filtering and filtering of reads without the appropriate RAD-tag was performed with Stacks' process_radtags.

**Read mapping and quality metrics.** The final libraries were mapped to the hg19 human reference genome (GRCh37_g1k) using BWA MEM (0.7.15)[32]. Resulting sam files were converted to bam, sorted and indexed using samtools (1.3.1)[33]. Quality metrics were calculated using GATK callableLoci (v3.7-0) for identifying loci with at least 10x coverage, Picard (2.9.0)[34] CollectInsertSizeMetrics to calculate fragment size histograms from mapped read pairs, and samtools flagstat to obtain mapping statistics.

**Somatic mutation calling.** Mutation calling was performed using GATK Mutect2[23], taking into account for the SNV metrics only reads with minimum mapping quality of 1, minimum base quality of 10 and excluding supplementary alignments, as well as discarding both reads in an overlapping read pair if they have different base calls at the locus of interest, or using just the read with highest base quality if they have the same base.

Additionally, Strelka (v 2.0.15) with disabled read depth filter was run on a subset of samples, taking into account for the SNV metrics only reads with minimum mapping quality of 1, minimum base quality of 10 and allowing a minimum alternate allele count of 2 and a minimum alternate allele frequency of 0.05 for a position to be considered in detecting SNV clusters.

For Mutect2- and Strelka-derived mutations, low-quality and spurious mutation calls were filtered by applying the following criteria[13]: VariantAlleleCountControl > 1, VariantMapQualMedian < 40.0, MapQualDiffMedian < -5.0 or MapQualDiffMedian > 5.0, LowMapQual > 0.05, VariantBaseQualMedian < 30.0, VariantAlleleCount > = 7 && VariantStrandBias < 0.05 && ReferenceStrandBias > = 0.2. The parameter ReadCountControl was set to be <20 for the three fresh-frozen and FFPE paired samples and <10 for the additional FFPE samples.

Additionally, based on the cosine similarity of WGS-derived mutational signatures and the mutational signatures derived for the initial three samples, we optimized the minimum number of reads supporting a SNV (fresh-frozen samples mutREAD = 5, WES = 7, 10× sWGS = 5, mutREAD FFPE = 10) and the minimal variant allele frequency of a SNV (fresh-frozen samples mutREAD = 0.03, WES = 0.01, 10× sWGS = 0.11, mutREAD FFPE = 0.13). The cut-offs were optimized separately for Strelka-derived mutations (fresh-frozen samples = 20 reads and 0.11 variant allele frequency, mutREAD FFPE = 11 and 0.03 variant allele frequency).

**Mutational signature profile.** The tri-nucleotide context for each SNV was determined using the SomaticSignatures R package[35]. Mutational signature profiles were derived for each sample using EAC-specific mutational signatures[13]. Finally, non-negative least squares in R was used to derive the contributions of each mutational signature to the overall mutational spectrum. The estimated coefficients were scaled to sum up to one.

**Reporting summary.** Further information on research design is available in the Nature Research Reporting Summary linked to this article.

## Data availability

All mutREAD data generated for the article will be available from European Genome-phenome Archive upon publication(accession number EGAD00001006170). WGS data for the matched patient samples is available from the ICGC data portal (https://dcc.icgc.org/, information about the patient ID are provided in Supplementary Table 8). The remaining data are available within the Article, Supplementary Information or available from the authors upon request.

## Code availability

All analysis code is freely available from https://github.com/jperner/mutREAD.

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

## Acknowledgements

The OCCAMS consortium for sample collection and sequencing was funded by a Programme Grant from Cancer Research UK. The laboratory of R.C.F. is funded by a Core Programme Grant from the Medical Research Council (RG84369). We thank the Human Research Tissue Bank, which is supported by the UK National Institute for Health Research (NIHR) Cambridge Biomedical Research Centre, from Addenbrooke's Hospital. Additional infrastructure support was provided from the Cancer Research UK–funded Experimental Cancer Medicine Centre. We thank Chris Laumer (EBI UK), the Genomics core at CRUK CI and the bioinformatics core at CRUK CI for valuable discussion and support. We would like to thank Dr. Maria O'Donovan, Dr. Shalini Malhotra and Dr. Ahmad Miremadi for histopathological assessment of samples.

## Author contributions

J.P., S.A., and K.N. designed experiments, interpreted the data, and wrote the paper. S.A. and K.N. performed experiments and J.P. performed computational analysis. G.D. conducted sequencing data management. M.E. contributed expertise for sequencing data analysis for this study. R.C.F. and S.T. supervised the work and helped to write the paper. R.F. obtained funding for the study and takes overall responsibility. All authors approved the final version of the paper.

## Competing interests

Rebecca Fitzgerald is named on patents related to the Cytosponge and associated assays which has been licensed by the MRC to Covidien (now Medtronic). RCF has performed consulting for Medtronic and is a shareholder in Cyted Ltd. Simon Tavaré is a member of the SAB of Ipsen, and a consultant for Kallyope Inc., New York. Rebecca Fitzgerald, Juliane Perner and Karol Nowicki-Osuch are named inventors on the patent filed for the mutREAD method by the University of Cambridge. The remaining authors declare no competing financial interests.
