## [Peer Review File · Nature Communications]

Reviewers' comments:

Reviewer #1 (Remarks to the Author): Expert in mutational signatures

The manuscript by Perner J et al describes an elegant, novel, cost effective and scalable method allowing extracting SNV-based mutational signatures from reduced representation of the genome material and from lower genome-wide coverage sequencing data. This interesting novel tool appears to have a lot of potential for use in more streamlined signature analyses in various settings, clinical, experimental etc.

The manuscript is well written, concise, and the techniques – laboratory ones as well as computational - are described clearly.

I have the following several comments:

Major:

1) It appears that the signature analysis relies on SNVs that were called by using only a single caller, MuTect2, which for various reasons might not be everyone's choice. Given the atypical nature of the input material due to reduced content and/or sample quality or low amounts, it would be interesting to compare the MuTect2 performance with other typical callers currently used/recommended for instance by the TCGA or ICGC consortia. Could the authors address this more systematically in the Manuscript by describing the consensus/union and unique outputs of 2-3 well-tested callers applied to the mutREAD data, and provide some recommendations/conclusions as to the impact of such multi-caller approach on the overall signature extraction results?

2) Most of the signatures seen in EACs (sig 1, 2/13, 17 and possibly 18 but not 3) have quite distinct and rather spiky profiles, and this could improve the reliability of their detection in the reduced content data. Could the authors test by for instance computational simulation (I am thinking using random selection of mutations corresponding to genome-wide coverage typically obtained by mutREAD) another data set that has more diverse signature content, for example a comparably sized liver HCC WGS data set from the ICGC data, to see if the reconstitution of the existing signature assignments and per-sample proportional distributions is as successful as seen in the EAC samples? Such exercise, if feasible, would possibly allow for making broader general statement about the mutREAD's utility and effectiveness.

Minor:

3) The concordance of the proportionate signature distribution between WGS and mutREAD within-sample shown in Figs 1D, 2A is impressive, attesting to the utility of this new approach. Unless I missed the information elsewhere in the manuscript, it would be helpful to also see how the matching techniques compare at the level of the SNV counts that are being compared. Could the authors include this information for each of the compared tumors, ideally by also showing in the same figures bar graphs based on the counts, or if this happens to be visually confusing due to for example displaying very high versus very low counts side by side, at least annotate the samples with the corresponding count numbers?

Reviewer #2 (Remarks to the Author): Expert in sequencing technologies and clinical genetics

In this study entitled "A cost-effective assay for quantifying mutational signatures in clinical cancer samples" by Perner et al, the authors presented the development of a sequencing and computational

method for the determination of mutational signatures in DNA samples. They showed their method, termed as mutREAD, mirrored the WGS-derived mutational signatures. Importantly, the method can be applied to DNA extracted from the FFPE samples with low tumor content. This work is meaningful with potential for applications.

The reviewer is not an expert on computational biology and thus would not comment on the computational technicality of the manuscript. But from the perspective of cancer genomics, the concerns are listed below:

1. The computational simulations were conducted using available data from WGS of 129 esophageal adenocarcinoma tumor samples. If the authors claim the method may have a wide range of applications, they will need to demonstrate it works not only on a type of solid tumor, but also on a type of leukemia and/or a type of lymphoma. For example, they may want to test mutREAD in lymphoma mutational signatures described in Chapuy, Nat Med, 24, 679, 2018.
2. Besides SNVs, does the method apply to copy number variations and structural abnormalities such as translocations and inversions etc?
3. If it is only demonstrated in the setting of esophageal adenocarcinoma, the scope of the ms would be quite limited and the ms may fit better to a journal with special focus on cancer computational genomics or a journal with special focus on clinical applicability such as Journal of Molecular Diagnostics.

Manuscript ID: NCOMMS-20-01433-A

We are grateful for the detailed and helpful reviewers' comments which we have addressed in a point by point fashion below.

Reviewers' comments:

Reviewer #1 (Remarks to the Author): Expert in mutational signatures

The manuscript by Perner J et al describes an elegant, novel, cost effective and scalable method allowing extracting SNV-based mutational signatures from reduced representation of the genome material and from lower genome-wide coverage sequencing data. This interesting novel tool appears to have a lot of potential for use in more streamlined signature analyses in various settings, clinical, experimental etc. The manuscript is well written, concise, and the techniques – laboratory ones as well as computational - are described clearly.

We would like to thank the reviewer for these comments.

I have the following several comments:

Major:

1) It appears that the signature analysis relies on SNVs that were called by using only a single caller, MuTect2, which for various reasons might not be everyone's choice. Given the atypical nature of the input material due to reduced content and/or sample quality or low amounts, it would be interesting to compare the MuTect2 performance with other typical callers currently used/recommended for instance by the TCGA or ICGC consortia. Could the authors address this more systematically in the Manuscript by describing the consensus/union and unique outputs of 2-3 well-tested callers applied to the mutREAD data, and provide some recommendations/conclusions as to the impact of such multi-caller approach on the overall signature extraction results?

Answer:

We agree with the reviewer that the choice of the mutation caller could have an impact on the mutational signatures. We have re-run the mutation calling for the three fresh-frozen and FFPE samples using Strelka, a commonly used variant caller, and compared the results to the Mutect2-derived mutations. For comparability, Mutect2 and Strelka were run with the same filtering parameters. Under this parameter setting, we detect more mutations with Mutect2 than with Strelka on the fresh-frozen mutREAD samples (Supplementary Table 3A, C). About 65-85% of the Strelka mutations are also detected with Mutect2. As expected, the variability of the mutation calls is higher on the FFPE mutREAD samples, with Mutect2 being more conservative in calling mutations.

The mutational signatures derived from both mutation callers and from the consensus mutation set were comparable and showed excellent cosine similarity to WGS-derived data (Supplementary Table 3B, D). This information is now included in Supplementary Table 3 and is referred to in the modified manuscript as follow:

Lines 107-109: “We observed similar cosine similarity between mutREAD and WGS when mutations were derived using an alternative mutation caller, Strelka (Supplementary Table 3).”

2) Most of the signatures seen in EACs (sig 1, 2/13, 17 and possibly 18 but not 3) have quite distinct and rather spiky profiles, and this could improve the reliability of their detection in the reduced content data. Could the authors test by for instance computational simulation (I am thinking using random selection of mutations corresponding to genome-wide coverage typically obtained by mutREAD) another data set that has more diverse signature content, for example a comparably sized liver HCC WGS data set from the ICGC data, to see if the reconstitution of the existing signature assignments and per-sample proportional distributions is as successful as seen in the EAC samples? Such exercise, if feasible, would possibly allow for making broader general statement about the mutREAD's utility and effectiveness.

Answer:

Thank you for this suggestion. We have performed a computational simulation using WGS data for different cancer types, including Liver-HCC, from the Pan-Cancer Analysis of Whole Genomes (PCAWG) network to check the performance of mutREAD. We have used the signature compendium from COSMIC to capture all mutational signatures relevant to the different cancer types. In this simulation, we achieved an average cosine similarity of 0.73 for Liver-HCC, which is among the top 5 cancer types in terms of cosine similarity. This data is now included in Supplementary Figure 1 and Supplementary File 1 and we have added the relevant text to the manuscript:

Lines 75-82: “We further investigated the applicability of RR-seq for estimating mutational signatures in different cancer types using the WGS data collected by the Pan-Cancer Analysis of Whole Genomes (PCAWG) network. RR-seq accurately estimated the mutational signature profiles across the majority of the 20 cancer types, including cancers with highly diverse mutational signature content, e.g. liver hepatocellular carcinoma (Liver HCC), and a non-solid tumor, i.e. B-cell non-Hodgkin lymphoma (Lymph-BNHL, Supplementary Figure 1A). As expected from our simulations above, the performance of the method was correlated with the mutational load across cancer types (Supplementary Figure 1B). Finally, RR-seq outperformed (expanded) WES in all cancer types (Supplementary File 1).”

Minor:

3) The concordance of the proportionate signature distribution between WGS and mutREAD within-sample shown in Figs 1D, 2A is impressive, attesting to the utility of this new approach. Unless I missed the information elsewhere in the manuscript, it would be helpful to also see how the matching techniques compare at the level of the SNV counts that are being compared. Could the authors include this information for each of the compared tumors, ideally by also showing in the same figures bar graphs based on the counts, or if this happens to be visually confusing due to for example displaying very high versus very low counts side by side, at least annotate the samples with the corresponding count numbers?

Answer:

We thank the reviewer for the suggestion. As the reviewer pointed out, it is difficult to plot the mutation counts of all methods side by side due to the large differences in magnitude. We have instead added this information to Supplementary table S2, S3, S4 and in Figure 2B, as well as the summary in the text:

Lines 103-105: “The mutational signatures, derived from 530-1471 mutations detected using GATK Mutect2, showed cosine similarities of 0.95-0.96 when compared with the WGS-derived mutational signature profiles (Figure 1D).”

Lines 113-114: “WES resulted in 46-325 mutations per sample and 10x sWGS identified 21-83 mutations per sample.”

Lines 121-122: “Cosine similarities to WGS-derived mutational signatures were between 0.89-0.96 based on 47-383 detected mutations.”

Reviewer #2 (Remarks to the Author): Expert in sequencing technologies and clinical genetics

In this study entitled “A cost-effective assay for quantifying mutational signatures in clinical cancer samples” by Perner et al, the authors presented the development of a sequencing and computational method for the determination of mutational signatures in DNA samples. They showed their method, termed as mutREAD, mirrored the WGS-derived mutational signatures. Importantly, the method can be applied to DNA extracted from the FFPE samples with low tumor content. This work is meaningful with potential for applications. The reviewer is not an expert on computational biology and thus would not comment on the computational technicality of the manuscript. But from the perspective of cancer genomics, the concerns are listed below:

We thank the reviewer for these comments.

1. The computational simulations were conducted using available data from WGS of 129 esophageal adenocarcinoma tumor samples. If the authors claim the method may have a wide range of applications, they will need to demonstrate it works not only on a type of solid tumor, but also on a type of leukemia and/or a type of lymphoma. For example, they may want to test mutREAD in lymphoma mutational signatures described in Chapuy, Nat Med, 24, 679, 2018.

Answer:

Thank you for this question which is similar to the point raised also by Reviewer #1. We have performed additional computational simulations to provide evidence for broad applicability of mutREAD. In terms of non-solid tumors, the PCAWG data set contains WGS for Lymph-BNHL, Myeloid-AML, Myeloid-MPN and Lymph-CLL with at least ten samples each.

Our computational simulation (Supplementary Figure 1, Supplementary File 1) shows that the best combination of restriction enzymes performed well for Lymph-BNHL (large B-Cell non-Hodgkin lymphoma) with a median cosine similarity of 0.73 and showed a median cosine similarity of 0.46 for Lymph-CLL (Chronic lymphocytic leukemia), 0.52 for Myeloid-AML (Acute myeloid leukemia) and 0.47 for Myeloid-MPN (myeloproliferative neoplasms). We noted that Lymph-BNHL samples have a higher mutational load (median number of mutations 7298) than the other three (Lymph-CLL=2286, Myeloid-AML=1305 and Myeloid-MPN=1067), possibly explaining the higher average cosine similarity for Lymph-BNHL. Hence, while performing well for the majority of cancer types, we cannot conclude that mutREAD will be applicable to all non-solid hematological malignancies. Nevertheless, mutREAD outperforms other available approaches, i.e. (expanded) WES, also for the above mentioned cancers (Supplementary File 1).

The following text has been added to the manuscript:

Lines 75-81: “We further investigated the applicability of RR-seq for estimating mutational signatures in different cancer types using the WGS data collected by the Pan-Cancer Analysis of Whole Genomes (PCAWG) network. RR-seq accurately estimated the mutational signature profiles across the majority of the 20 cancer types, including cancers with highly diverse mutational signature content, e.g. liver hepatocellular carcinoma (Liver HCC), and a non-solid tumor, i.e. B-cell non-Hodgkin lymphoma (Lymph-BNHL, Supplementary Figure 1A). As expected from our simulations above, the performance of the method was correlated with the mutational load across cancer types (Supplementary Figure 1B). Finally, RR-seq outperformed (expanded) WES in all cancer types (Supplementary File 1).”

2. Besides SNVs, does the method apply to copy number variations and structural abnormalities such as translocations and inversions etc?

Answer:

This is a very good question and something that we have been looking into since if mutREAD could be used to infer copy number variations this would offer a further range of applications. As mentioned in the discussion (lines 152-153), RR-seq based methods have been already applied in this context¹. We thus anticipate that mutREAD can also be used to infer copy number variations. However, we would prefer to perform further analysis and validation than include it in the current version of the manuscript.

Structural variant (SV) calling relies on the identification of DNA breakpoints via the detection of split reads and on read pair orientation. The PCAWG analysis identified a median number of ~170 breakpoints per EAC case, one of the most strongly SV-affected tumor types². Since mutREAD samples less than 1% of the human genome, the chances of covering a break point are low and hence, traditional SV calling methods will likely not be able applicable.

3. If it is only demonstrated in the setting of esophageal adenocarcinoma, the scope of the ms would be quite limited and the ms may fit better to a journal with special focus on cancer computational genomics or a journal with special focus on clinical applicability such as Journal of Molecular Diagnostics.

Answer:

Given the results of the computational simulations on a large number of different cancer types, we believe that mutREAD will offer a useful additional tool to study mutational signatures for the majority of cancer types. The range of cancer types tested includes highly heterogeneous cancer types, such as liver hepatocellular carcinoma, and a non-solid tumor, i.e. BNHL, for which mutREAD performed well. With this additional information, we feel that mutREAD has broad appeal to cancer research community. mutREAD's compatibility with FFPE material and the low per-sample cost allows the clinical research community to analyze a larger number of samples for mutational signatures. We therefore anticipate that our revised manuscript will be of interest to readers of Nature Communications.

1. Zheng, C. *et al.* Determination of genomic copy number alteration emphasizing a restriction site-based strategy of genome re-sequencing. *Bioinformatics* **29**, 2813–2821 (2013).
2. Li, Y. *et al.* Patterns of somatic structural variation in human cancer genomes. *Nature* **578**, 112–121 (2020).

REVIEWERS' COMMENTS:

Reviewer #1 (Remarks to the Author):

Overall, the authors addressed the main concerns and significantly improved the manuscript. I have only a couple of minor points to mention:

1) It is satisfactory that the signatures and cosine similarities were comparable to the previous observations, after including Strelka as an alternative variant caller. However, there still seems an issue that merits more discussion, and that is the effects of the individual callers on the obtained mutation counts and on their overlap. Specifically, the FF MutREAD consensus (Supp Table 3A) represents on average a mere 44% of the MuTect calls (this drop is not observed by WGS), and in Supp Table 3C (FFPE mutation counts) there is about the same average drop seen (44%). There might not be much to do about these differences and the performance of these specific callers on mutREAD data. However, the choice of callers, their performance settings and the effects on the resulting mutation counts should be discussed in more depth, simply to raise the awareness of the readers about a caveat to be systematically addressed through benchmarking and optimization.

2) What is now Suppl Table 4 is labeled on the xls sheet Supp Table 3, while the new Supp Table 3 (the caller comparison) has no sheet labeling within – please check if all referencing of display/data items resulting from the revision

Reviewer #2 (Remarks to the Author):

The responses to the reviewer's concerns are adequate. No further comments.

REVIEWERS' COMMENTS:

Reviewer #1 (Remarks to the Author):

Overall, the authors addressed the main concerns and significantly improved the manuscript. I have only a couple of minor points to mention:

1) It is satisfactory that the signatures and cosine similarities were comparable to the previous observations, after including Strelka as an alternative variant caller. However, there still seems an issue that merits more discussion, and that is the effects of the individual callers on the obtained mutation counts and on their overlap. Specifically, the FF MutREAD consensus (Supp Table 3A) represents on average a mere 44% of the MuTect calls (this drop is not observed by WGS), and in Supp Table 3C (FFPE mutation counts) there is about the same average drop seen (44%). There might not be much to do about these differences and the performance of these specific callers on mutREAD data. However, the choice of callers, their performance settings and the effects on the resulting mutation counts should be discussed in more depth, simply to raise the awareness of the readers about a caveat to be systematically addressed through benchmarking and optimization.

We thank the reviewer for raising this important point. We have added the discussion to the text: “We note that the two mutation callers agree in 41-85% of mutations, suggesting that for optimal sensitivity dedicated benchmarking and optimization is necessary when applying different mutation callers.”

2) What is now Suppl Table 4 is labeled on the xls sheet Supp Table 3, while the new Supp Table 3 (the caller comparison) has no sheet labeling within – please check if all referencing of display/data items resulting from the revision

We thank the reviewer for raising this point. We have now corrected the names of sheets on the supplementary tables. We have also checked all the reference to the tables and they are correct.

Reviewer #2 (Remarks to the Author):

The responses to the reviewer's concerns are adequate. No further comments.